# The Impact of the 2019 European Guideline for Cardiovascular Risk Management: A Cross-Sectional Study in General Practice

**DOI:** 10.3390/jcm9072140

**Published:** 2020-07-07

**Authors:** Rahel Meier, Yael Rachamin, Thomas Rosemann, Stefan Markun

**Affiliations:** Institute of Primary Care, University of Zurich and University Hospital Zurich, Pestalozzistr. 24, 8091 Zurich, Switzerland; yael.rachamin@usz.ch (Y.R.); thomas.rosemann@usz.ch (T.R.); stefan.markun@usz.ch (S.M.)

**Keywords:** cardiovascular risk management, general practice, guideline, lipid-lowering treatment

## Abstract

The aim of this study was to assess the impact of the 2019 published European Society of Cardiology (ESC)/European Atherosclerosis Society (EAS) guideline on cardiovascular (CV) risk management compared with its predecessor from 2016 in a cohort in general practice. We performed a cross-sectional retrospective study with data from electronic medical records. The study cohort included 103,351 patients with known CV risk. We assessed changes in CV risk classification and low-density lipoprotein cholesterol (LDL-C) target values, the impact on LDL-C achievement rates, and the current lipid-lowering treatments. Under the 2019 ESC guideline, CV risk categories changed in 27.5% of patients, LDL-C target levels decreased in 71.4% of patients, and LDL-C target achievement rate dropped from 31.1% to 16.5%. Among non-achievers according to the 2019 guideline, 52.2% lacked lipid-lowering drugs entirely, and 41.5% had conventional drugs at a submaximal intensity. Of patients in the high-risk and very high-risk categories, at least 5% failed to achieve the LDL-C target level despite treatment at maximal intensity with conventional lipid-lowering drugs, making them eligible for PCSK-9 inhibitors. In conclusion, the 2019 ESC/EAS guideline lowered LDL-C target values for the majority of patients in general practice and halved LDL-C target achievement rates. There is still a large undeveloped potential to lower CV risk by introducing conventional lipid-lowering drugs, particularly in patients at high or very high CV risk. A substantial proportion of the patients can only achieve their LDL-C targets using PCSK-9 inhibitors, which would currently require an at least 10-fold increase in prescribing of these drugs.

## 1. Introduction

Cardiovascular (CV) disease is the leading cause of death in Europe, accounting for 45% of all deaths [1]. Multiple risk factors contribute to CV disease [2], and many are preventable or treatable, including hypertension or elevated levels of low-density lipoprotein cholesterol (LDL-C) [3]. Individual risk factors are of variable importance and their individual contribution to overall CV risk is complex. Risk stratification schemes are widely used to reduce complexity in risk estimation for individual patients. They typically include morbidities such as hypertension or diabetes mellitus (DM) and laboratory values such as total cholesterol [2].

In Europe, the most widely used risk classification scheme is the one proposed by the European Society of Cardiology (ESC) and the European Atherosclerosis Society (EAS). It divides patients into four risk categories ranging from “low risk” to “very high risk” [4,5]. To lower CV risk, the ESC/EAS specifies LDL-C target values for each risk category and, where required, recommends pharmacological treatment to achieve those target values. In August 2019, a major update of the ESC/EAS guideline for the management of dyslipidemia was published, with changes to the CV risk classification scheme and LDL-C target values [4]. The revised CV risk classification scheme includes adaptations to the way morbidities and laboratory values are accounted for. Furthermore, LDL-C target values were lowered for most risk categories. The 2019 update on the ESC/EAS guideline is substantial, and it necessitates risk classification as well as LDL-C target value to be updated in certain patients. Given its complex nature and the interplay of factors relevant to risk classification, the proportion of patients actually requiring a change in risk classification and LDL-C target value in general practice is uncertain. Furthermore, it is unclear how the proportions of patients achieving LDL-C target values will change compared to the 2016 guideline, and what the related therapeutic implications are for patients in real-life general practice.

Therefore, the aim of this study was to assess the impact of the 2019 ESC/EAS guidelines on CV risk management in a cohort of patients in real-life general practice. The specific objectives of this study were to evaluate (1) the proportions of patients for whom the 2019 ESC/EAS guideline entailed changes in CV risk classification and LDL-C target values, together with the directions (increase or decrease) and reasons for reclassification. (2) The impact on LDL-C target achievement rates for each CV risk category. (3) Current lipid-lowering treatments (in terms of dosage and drug combinations) of patients with available LDL-C measurements but failing to achieve the 2019 target levels, stratified by 2019 CV risk category.

## 2. Methods

### 2.1. Study Design, Setting, and Participants

We performed a cross-sectional retrospective analysis using data from the family medicine international classification of primary care (ICPC) research using electronic medical records (FIRE) project [6]. As of August 2019, more than 540 participating Swiss general practitioners (GPs), i.e., 10.5% of GPs working in the German-speaking region of Switzerland [7], provided anonymized patient and routine data from their electronic medical records to the FIRE database. We included all patients with at least one consultation in the study observation period starting 1 September 2016 (publication of the 2016 ESC guideline), and ending 31 August 2019 (publication of the 2019 ESC/EAS guideline) (Figure 1). Patients were eligible if available data allowed for CV risk classification according to the 2016 or 2019 ESC/EAS guideline (see below). The local Ethics Committee of the Canton of Zurich waived approval, because the FIRE project was considered outside the scope of the Federal Act on Research involving Human Beings (BASEC-Nr. Req-2017-00797).

### 2.2. Implementation of CV Risk Classification

Both the 2016 and 2019 ESC/EAS guidelines combine two different assessments to define the CV risk categories, the first of which being the “Systematic COronary Risk Estimation” (SCORE) [8], and the second reflecting the presence of specific morbidities and risk factors. Here we provide an overview of the classification process, which was implemented within the FIRE database to assess the CV risk categories for each patient according to the 2016 and 2019 ESC/EAS guidelines (see Appendix A for the fully detailed description of the classification process).

Risk assessment using the SCORE directly provides prognostic probabilities for fatal CV events based on gender, age (40–70 years), smoking status, parametric values of systolic blood pressure (120–180 mmHg) and total cholesterol (4–8 mmol/L) in untreated patients without CV disease. We calculated SCORE probabilities according to the guidelines’ instructions for low-risk countries. We used the most recent untreated total cholesterol and systolic blood pressure values if they were concurrently available within 5 years as CV risk progression is considered to be small during the length of this period [9].

The risk assessment based on morbidity and risk factors encompasses established CV disease, DM with target organ damage, DM with major risk factors (advanced age, smoker, dyslipidemia, hypertension, and obesity) [10], severe or moderate chronic kidney disease (CKD), markedly elevated single risk factors, and DM without risk factors/target damage. We identified the above using ICPC-2 codes [11], laboratory values, vital signs, or anatomical therapeutic chemical [12] codes for medication exclusively indicated for the abovementioned morbidities depending on availability. No age restrictions were applied.

We combined the SCORE and the risk classifications based on morbidity to generate the composite ESC/EAS risk classification. Whenever multiple reasons for classification were available at the same time, the one with the higher CV risk was adopted. A detailed analysis highlighting differences between the 2016 and 2019 ESC/EAS guidelines is provided in the Appendix A.

### 2.3. Database Query and Variables

From the FIRE database, we extracted the following patient data: gender; year of birth; most recent CV risk category within 10 years according to both the 2016 and 2019 ESC/EAS guidelines as described above; reasons for classification according to the 2016 and 2019 ESC/EAS guidelines, respectively (specific morbidity or SCORE); value of the last available LDL-C measure in the observation period after the last CV risk assessment; information (product, daily dose) about lipid-lowering drugs (statins, ezetimibe, proprotein convertase subtilisin/kexin-9 (PCSK-9) inhibitors, and their combinations). The intensity of statin treatment was inferred from drug name and daily dose according to the classification in the 2014 American College of Cardiology and American Heart Association guidelines [13].

### 2.4. Data Analysis

LDL-C target values were adopted as reported in the ESC/EAS guidelines. In 2016, target values were <3.0 mmol/L for low/moderate risk, <2.6 mmol/L for high risk, and <1.8 mmol/L for very high risk; in 2019 target values were <3.0 mmol/L for low risk, <2.6 mmol/L for moderate risk, <1.8 mmol/L for high risk, and <1.4 mmol/L for very high risk.

We carried out all analyses using the statistical software package R (Version 3.5.0) [14]. We used counts and proportions (*n* and %) as well as medians with interquartile ranges (IQR) to describe the data.

## 3. Results

### 3.1. Characteristics of Patients

We assessed half a million patients in general practice and identified 103,351 with known CV risk and thus eligible for this study (Figure 1). The patients’ median age at the end of the observation period was 64 years (IQR = 53–76), and 49.2% (*n* = 50,884) were female. LDL-C could be followed up in 23.6% (*n* = 24,356) of patients after their CV risk was determined. The distribution across the four risk categories according to the 2016 ESC guideline was as follows: low risk, 9.6%; moderate risk, 21.4%; high risk, 29.5%; and very high risk, 39.6%. Based on the 2019 ESC/EAS guideline, the distribution was as follows: low risk, 9.8%; moderate risk, 17.0%; high risk, 53.1%; and very high risk, 20.1%. In the low- and moderate-risk categories, all patients were identified via their SCORE values whereas in the high- and very high-risk categories, only a minor percentage was identified by SCORE values (high-risk category 2016: 6.3%, 2019: 8.2%; very high-risk category 2016: 0.2%, 2019: 6.3%). Detailed patient characteristics stratified by guideline and risk category are presented in Table 1.

### 3.2. Impact of Guideline Update on Risk Classification and LDL-C Target Values

The 2019 ESC/EAS guideline caused a change in CV risk classification in 27.5% (*n* = 28,419) of patients. Specifically, the risk category decreased in 19.8% (*n* = 20,493) and increased in 3.4% (*n* = 3507). In addition, 4.3% (*n* = 4419) were newly classified (i.e., without classification under the criteria of the 2016 ESC guideline). The reasons for risk category reclassification or new classification were modifications to the identification scheme for DM with major risk factors (18.8%, *n* = 19,422), SCORE adaptations (5.2%, *n* = 5354), and adaptation in the identification scheme of markedly elevated single risk factors (3.5%, *n* = 3643). The changes to the identification scheme for DM with major risk factors led to downgrading risk in patients with DM with only one or two major risk factors from the very high- to the high-risk category. For a detailed visualization of the reasons for risk categories reclassification, see the Appendix A.

LDL-C target values changed in 71.4% (*n* = 73,781) of patients. All changes to LDL-C targets resulted in lower LDL-C target values. The impact of the 2019 ESC/EAS guideline on risk classification and LDL-C target levels is shown in Figure 2. The median LDL-C distance to target level increased in the moderate-risk category by a factor of 2 (2016: 0.3 (0–1.0) mmol/L; 2019: 0.6 (0–1.2) mmol/L), in the high-risk category by a factor of 2.8 (2016: 0.4 (0–1.3) mmol/L; 2019: 1.1 (0.3–1.9) mmol/L), and in the very high-risk category by a factor of 1.6 (2016: 0.5 (0–1.3) mmol/L; 2019: 0.8 (0.3–1.6) mmol/L). No changes in LDL-C target values were introduced in the low-risk category.

### 3.3. Impact of Guideline Update on LDL-C Target Value Achievement

A follow-up LDL-C value needed to assess target achievement was available in 24.6% (*n* = 24,356) of patients classified according to the 2016 ESC guideline and in 23.9% (*n* = 24,670) of patients classified according to the 2019 ESC/EAS guideline. In total, 31.1% (*n* = 7582) of patients achieved the recommended LDL-C target value according to the 2016 ESC guideline, and 16.5% (*n* = 4066) according to the 2019 ESC/EAS guideline. Figure 3 shows the 2016 and 2019 target achievement rates stratified by risk category.

### 3.4. Lipid-Lowering Treatment in LDL-C Target Non-Achievers

Of the patients not achieving LDL-C target values recommended by the 2019 ESC/EAS guideline (*n* = 20,604), 52.2% (*n* = 10,748) received no lipid-lowering drugs at all, 41.5% (*n* = 8550) were treated with statins only, 5.5% (*n* = 1139) received a combination of statins and ezetimibe, and 0.11% (*n* = 22) received a statin and a PCSK-9 inhibitor. Of the patients treated with statins, 38.5% (*n* = 3730) received a high-intensity treatment. Detailed numbers stratified by risk category are given in Table 2.

## 4. Discussion

In this study, we assessed the impact of the new 2019 ESC/EAS guideline on CV risk management in a cohort in general practice with known CV risk. The new guideline’s impact was extensive and lowered LDL-C target values for 71% of the patients. In the most relevant group of patients in the high-risk and very high-risk categories, only 15% of patients currently achieve their respective LDL-C target values according to the 2019 guideline, suggesting that intensified treatment is needed. While in most of these patients, conventional lipid-lowering drug treatment can either be initiated or intensified, in at least 5%, the conventional treatment methods are exhausted and PCSK-9 inhibitors are recommended. In practice, this would translate into an over 10-fold increase in the prescription of PCSK-9 inhibitors.

With respect to its predecessor from 2016, changes in the 2019 ESC/EAS guideline comprised updated classification criteria and new LDL-C target values for CV risk categories. When applied to our real-life general practice cohort, we noted few changes in classification criteria for low-risk and moderate-risk categories, but every second patient in the former very high-risk category was downgraded into the high-risk category. Thereby, the guideline change caused a net downgrading in CV risk. However, it is important to understand that by also lowering the LDL-C target levels across all but the low-risk category, the new guideline actually tightened the recommendations for 71% of all patients. For clinicians, the identification of patients with changed LDL-C target levels is straightforward: LDL-C target levels changed for everyone except for patients with DM and less than three additional CV risk factors, and for those in the low-risk category.

By tightening treatment goals, the new 2019 ESC/EAS guideline increased the proportion of patients not meeting their LDL-C targets. In patients in the high-risk and very high-risk categories, the LDL-C target achievement rate halved, from 30% to 15%. This population may be the most relevant to consider, since evidence of the effectiveness of lipid-lowering drugs is weaker in low-risk and moderate-risk patients [15,16]. With respect to the current treatment of the patients who are not achieving their LDL-C target values, we found that half the patients still used no lipid-lowering drugs at all. The largest undeveloped potential to reach LDL-C target values therefore lies in initiating lipid-lowering treatment in the first place. This finding, surprising as it may seem, is in line with study results covering several other major European healthcare systems [17,18,19]. The second largest potential lies with the 45% of patients receiving conventional lipid-lowering drug treatment at submaximal intensity. Side effects, however, might limit maximizing the intensity of treatment in several of these cases [20]. At least 5% of the patients categorized as high-risk or very high-risk receive maximum lipid-lowering treatment already, and could further approximate their LDL-C target values only by introducing a PCSK-9 inhibitor. The true proportion benefitting from PCSK-9 inhibitors might be even higher as we were unable to account for statin intolerance. Within our cohort in general practice, the current prescription rate for PCSK-9 inhibitors is 0.37% and it would thus need to increase more than 10-fold to achieve LDL-C targets in these patients at high or very high CV risk. Given the current prices of PCSK-9 inhibitors and the small additional benefits in terms of absolute risk reductions, the implementation of such a recommendation may be contested from a cost-efficiency perspective [21,22,23].

Economic barriers may however not be the bottleneck to overcome in order to fully adopt the current guideline in general practice. Implementation of clinical practice guidelines is often slow in general practice, and guidelines on CV risk reduction are no exception [24,25,26]. Since 1987, guidelines on CV risk reduction have gradually increased the proportion of patients eligible for lipid-lowering drugs [27], and the current 2019 ESC/EAS guideline continues this trend. In general practice, such changes often encounter disagreement and lack of applicability for various reasons [28]. Given these multiple barriers in the health care chain, coordinated national strategies may be required to tailor and successfully implement recommended changes in CV risk management [29].

### Strength and Limitations

To our knowledge, this is the first study assessing the real-life impact of the new 2019 ESC/EAS guideline on CV risk management. Our study encompassed over 100,000 individual patients from a general practice cohort, and is therefore highly representative of the large population to which the new guideline applies.

The major limitation of this study is its inherent risk of selection bias on different levels. Selection bias on the GP level might occur since participation in the FIRE project is voluntary and requires the use of electronic medical records which is not standard in Switzerland [30]. Therefore, GPs contributing to the FIRE database might represent a higher performing sample of GPs. However, age and gender structure of GPs participating in the FIRE project is similar to census data published by the Swiss medical association [31]. Furthermore, there is a risk for information bias: failure to enter data creates a risk of under-estimating CV risk categories and current drug treatments, or might even prevent assessing CV risk categories at all. CV risk categories may therefore be systematically under-estimated, and our results should be understood as minimal estimates. Laboratory values, however, are automatically fed into the database; therefore, our results regarding LDL-C target achievement are likely robust. However, it is still likely that the selection of patients with available LDL-C measurements entailed a certain risk of bias, as these patients might be more closely managed due to their higher needs. Additionally, it should be noted that the medication information stems from prescriptions, and information on patient compliance was not available. Regarding ESC/EAS guideline implementation, we noted that the guidelines leave room for interpretation. For example, there is some ambiguity in assigning SCORE values for certain parameter ranges (age, LDL-C and blood pressure). This required us to make more precise definitions than actually stated in the guideline. Additionally, we extended age categories for the 2016 ESC guideline to range from 65 to 67 to achieve comparability with the 2019 ESC/EAS guideline. Thus, the results reflect, to a small degree, our own interpretation of the guidelines, but the same room for interpretations is also left to GPs when applying it. Lastly, our definition of LDL-C target level achievement rests exclusively on reaching the respective thresholds, whereas a 50% reduction in LDL-C also qualifies as target achievement according to the guidelines. We were unable to determine this measure because pre-treatment LDL-C values were only available for a minority of patients. Such electronic medical record-specific limitations, however, are not a relevant limitation to reliability in predicting CV risk, according to Wolfson et al. [32].

## 5. Conclusions

In conclusion, the impact of the 2019 update of the ESC/EAS guideline for the management of dyslipidemia is substantial, lowering the LDL-C target values for 71% of patients with known CV risk in general practice. Achievement rates of LDL-C targets were halved and increased the proportions of patients eligible for intensified lipid-lowering treatment. While in most cases initiating or increasing intensity of conventional lipid-lowering treatment is recommended, at least 5% of patients are eligible for PCSK-9 inhibitors, which would lead to a 10-fold increase of prescriptions for these drugs.

## Figures and Tables

**Figure 1 jcm-09-02140-f001:**
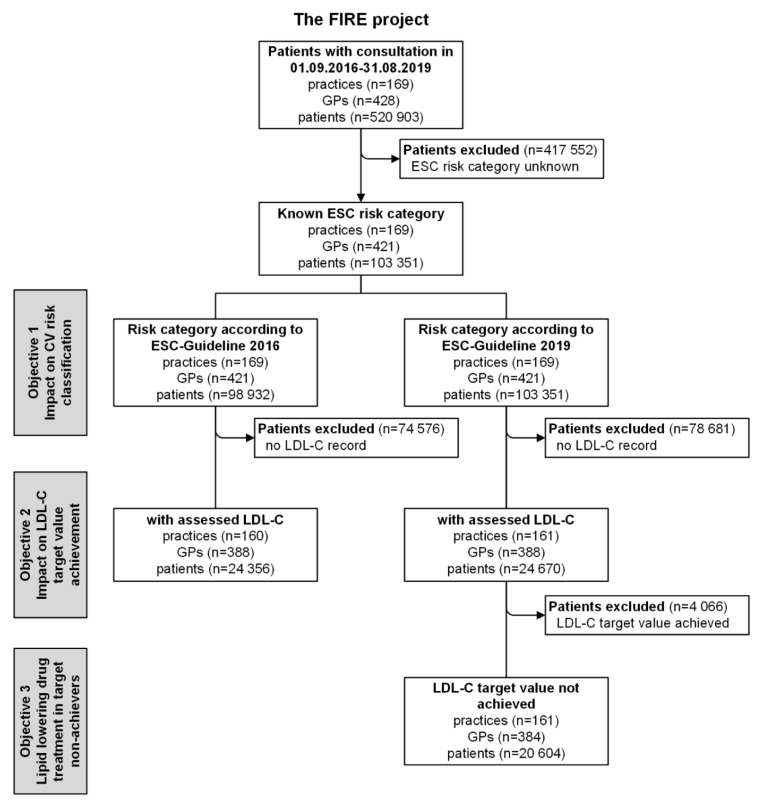
Flowchart of study protocol. GPs: general practitioners; ESC: European Society of Cardiology; LDL-C: low-density lipoprotein cholesterol.

**Figure 2 jcm-09-02140-f002:**
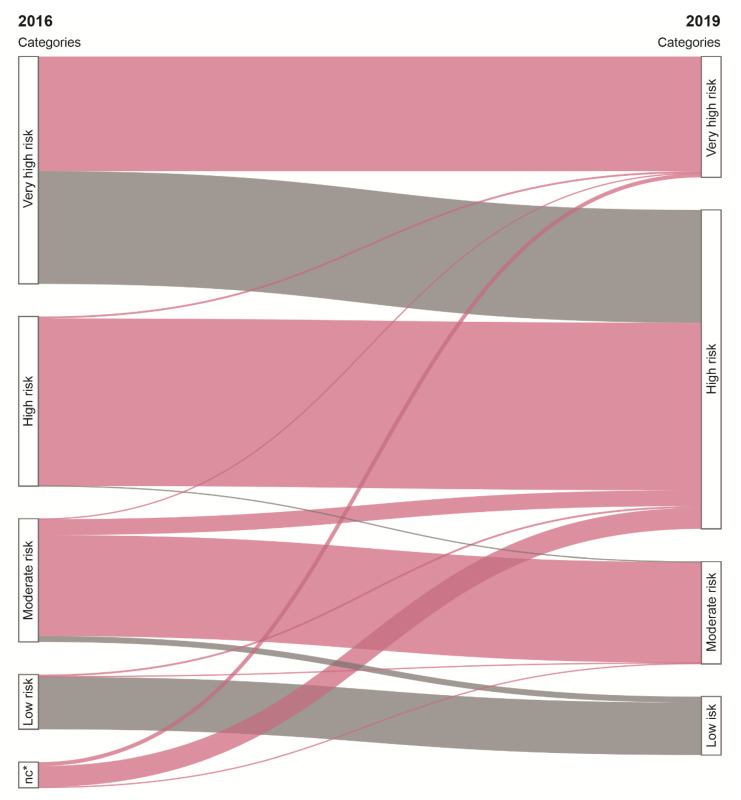
Impact of the 2019 ESC/EAS guideline on CV risk classification and LDL-C target values. Flows represent patients’ classification according to the 2016 and 2019 guidelines; the size of each flow is in proportion to the number of patients. Colors indicate changes in LDL-C target values: red: decrease of LDL-C target value, grey: no change of LDL-C target value. * no classification.

**Figure 3 jcm-09-02140-f003:**
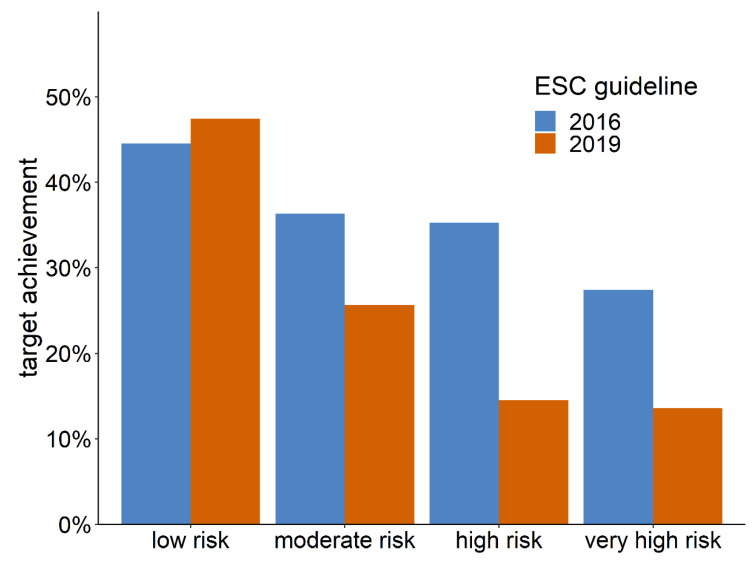
LDL-C target achievement stratified by guideline and risk category.

**Table 1 jcm-09-02140-t001:** Patient characteristics stratified according to ESC/European Atherosclerosis Society (EAS) guideline and cardiovascular (CV) risk category (total number of patients in 2016: 98,932; total number of patients in 2019: 103,351).

2016 Guideline				
Patient Characteristics	Low Risk	Moderate Risk	High Risk	Very High Risk
	(*n* = 9461)	(*n* = 21,138)	(*n* = 29,176)	(*n* = 39,157)
Median age (IQR)	47 (44–51)	58 (53–63)	69 (53–81)	72 (61–81)
% female	75.7	38.6	57.0	42.6
% with an LDL-C measurement	9.9	16.1	19.1	36.9
median LDL-C (IQR) mmol/L	3.1 (2.5–3.8)	3.3 (2.7–4)	3.0 (2.3–3.9)	2.3 (1.8–3.1)
Morbidities				
s % with previous CVD	0.0	0.0	0.0	27.8
% with severe CKD	0.0	0.0	0.0	10.2
% with moderate CKD	0.0	0.0	57.9	25.1
% with diabetes	0.0	0.0	23.6	74.2
% with dyslipidemia	53.5	68.7	32.5	39.8
% with hypertension	11.7	22.3	43.4	67.2
% with obesity	15.7	16.3	12.5	25.5
Lipid-lowering drugs				
% no treatment	97.6	93.3	80.0	52.2
% statin only	2.1	6.0	18.3	42.7
% statin and ezetimibe	0.18	0.46	1.37	4.52
% ezetimibe only	0.11	0.19	0.33	0.49
% statin and PCSK-9 inhibitors	0.00	0.00	0.03	0.06
% PCSK-9 inhibitors only	0.00	0.00	0.01	0.02
**2019 Guideline**				
**Patient Characteristics**	**Low Risk**	**Moderate Risk**	**High Risk**	**Very High Risk**
	**(*n* = 10,094)**	**(*n* = 17,583)**	**(*n* = 54,876)**	**(*n* = 20,798)**
Median age (IQR)	48 (44–52)	58 (53–62)	68 (56–78)	74 (66–83)
% female	74.8	38.9	51.8	38.7
% with an LDL-C measurement	9.0	13.5	24.6	37.9
median LDL-C (IQR) mmol/L	3.1 (2.5–3.6)	3.2 (2.6–3.8)	2.9 (2.1–3.7)	2.2 (1.7–3)
Morbidities				
% with previous CVD	0.0	0.0	0.0	52.4
% with severe CKD	0.0	0.0	0.0	19.1
% with moderate CKD	0.0	0.0	36.8	31.5
% with diabetes	0.0	0.0	47.9	46.4
% with dyslipidemia	52.2	67.0	38.5	48.1
% with hypertension	10.8	22.7	45.4	78.6
% with obesity	14.9	16.9	15.4	29.8
Lipid-lowering drugs				
% no treatment	98.0	94.2	74.6	42.2
% statin only	1.8	5.3	23.4	50.5
% statin and ezetimibe	0.11	0.39	1.61	6.63
% ezetimibe only	0.09	0.15	0.38	0.60
% statin and PCSK-9 inhibitors	0.00	0.00	0.02	0.12
% PCSK-9 inhibitors only	0.00	0.01	0.01	0.04

IQR: interquartile range; LDL-C: low-density lipoprotein cholesterol; CVD: cardiovascular disease; CKD: chronic kidney disease; PCSK-9: proprotein convertase subtilisin/kexin-9.

**Table 2 jcm-09-02140-t002:** Characteristics of patients not achieving LDL-C target values according to the 2019 ESC/EAS guideline (total number of patients = 20,604).

2019				
Patient Characteristics	Low Risk	Moderate Risk	High Risk	Very High Risk
	(*n* = 475)	(*n* = 1769)	(*n* = 11,551)	(*n* = 6809)
Median age (IQR)	49 (45–53)	59 (55–63)	67 (57–76)	72 (64–79)
% female	81.3	41.0	51.5	35.5
median LDL-C (IQR) in mmol/L	3.6 (3.3–4.0)	3.5 (3.1–4.0)	3.1 (2.5–3.9)	2.4 (1.9–3.2)
Median distance to LDL-C target (IQR) in mmol/L	0.6 (0.3–1.0)	0.9 (0.5–1.4)	1.3 (0.7–2.1)	1.0 (0.5–1.8)
Lipid-lowering drugs				
% no treatment	93.3	89.1	58.5	28.9
% statin only	5.3	10.1	37.7	58.7
% statin and ezetimibe	0.84	0.45	3.05	11.4
% ezetimibe only	0.63	0.40	0.74	0.84
% statin and PCSK-9 inhibitors	0.00	0.00	0.06	0.22
% PCSK-9 inhibitors only	0.00	0.00	0.02	0.07
Statin treatment intensity				
% high	1.5	2.3	12.0	33.8
% moderate	2.9	6.9	23.7	29.9
% low	0.2	0.5	1.7	2.1
% missing	1.5	0.9	3.3	4.4

IQR: interquartile range; LDL-C: low-density lipoprotein cholesterol; PCSK-9: proprotein convertase subtilisin/kexin-9.

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
