# Peer review of "The Impact of the 2019 European Guideline for Cardiovascular Risk Management: A Cross-Sectional Study in General Practice"

_jcm, 2020, doi:10.3390/jcm9072140_

Round 1

Reviewer 1 Report

The paper is focused on a very interesting topic.

Please improve the introduction, the discussion and describe the methods more adequately.

In addition there is a moderate English change required.

Author Response

Dear Reviewer

Thank you for consideration and the opportunity to revise our manuscript. We highly appreciate your feedback.

Our point-by-point response to the reviewer's comments are included in the attachment. Page and line numbers refer to the revised manuscript file with tracked changes. The bibliography has been updated accordingly.

We hope you find our responses satisfactory.

Sincerely,

Rahel Meier

Reviewer 2 Report

In this interesting study, the authors analyze the impact of the 2019 published European Society of Cardiology (ESC) guideline on CV risk management in comparison with the 2016 guideline in a cohort of initially 520 903 subjects attended in general practice. Although the proportion of decrease in estimated CV risk category (19.8%) was six-fold the increase (3.4%), LDL-C target values changed in over 70% of cases and exclusively towards a lower threshold, thereby halving the rate of accomplishment from 31.1% to 16.5%.

Several aspects raise some concern and should be addressed.

  1. Quality of the database

The high proportion of excluded patients indicates a poor quality of the electronic medical records. Globally, 520 903 subjects were eligible for the study, but over 80% of them were excluded for the first objective because “ESC risk category unknown”. If I understood well, even making the assumptions in the supplementary material, ESC risk could not be calculated in any way, indicating that data introduction by GPs was poor.

In a further step, due to missing LDL-C values, another over 70 000 patients were excluded to answer the second objective. At the end, less than 5% of the eligible patients provided data to draw the conclusions. Considering that only 10.5% of GPs participated in the study, adds to the concern about the quality of the database.

  1. Estimation of CV risk

The authors describe a so-called “multistep identification process” in order to define CV risk because, as a matter of fact, the 2019 Guideline recommends the SCORE tables but also uses the combination of comorbidities and risk factors to classify risk. The following two observations deserve some comment by the authors:

As mentioned earlier, over 70 000 had no LDL-C record, so it must be assumed, that CV risk estimation in these patients derived only from morbidity based risk identification. This fact enhances the lack of consistency of the database, for introducing diagnostic codes is less solid than laboratory values. Did the authors compare the results of the multistep identification process with the pure SCORE classification?

Besides, the authors decided that “cholesterol values and systolic blood pressure values were valid for a maximum of 5 years.” Were missing values the reason for this assumption? Was any sensitivity analysis performed to check if validity had been defined for a different period of time, for example, 2 years? What was the proportion of patients having all necessary date simultaneously to classify them?

  1. Compliance with the medication

One important issue in describing CV risk related with electronic medical records is the method to check compliance with medication. Introduction of ATC codes means prescription but not automatically compliance. Have the authors access to information concerning dispensation of drugs, that should better reflect compliance than prescription?

  1. Factors modifying Systematic Coronary Risk Estimation risk

Left ventricular hypertrophy is considered an important factor modifying risk estimation (see Box 4 in the 2019 Guideline). In fact, it is the most prevalent target organ damage in hypertensive patients, but it is not mentioned in this study. The authors might comment on the reason for excluding this important factor.

  1. Treatment of older people

The authors identify subjects without statins or other lipid lowering drugs as the most susceptible group to improve LDL-C target achievements by initiating therapy. In fact, and according to Table 1, 42.2% of subjects of very high risk did not have any lipid lowering drugs prescribed. But median age in this group was 74 years. And again, in Table 2, showing patients not achieving LDL-C target values, 28.9% of patients in the very high risk group were not under lipid lowering therapy. The median age in this group was 72 years.

These data give the impression that reducing CV risk in this study should be carried out at the cost of otherwise healthy subjects who due to their advanced age are classified as very high risk. To clarify this item, perhaps the authors could specify how many of these very old subject fall under the category of primary prevention? The answer to this question is important as evidence for primary prevention in the very old is very controversial (class IIb, level B).

  1. Use of PCSK-9 inhibitors

One of the main conclusions of this study is the eligibility of 5% of the patients for using PCSK-9 inhibitors on the basis of possible side effects of statins. Did the authors have any estimate of statin-intolerance in the study population? Without such a figure, proposing PCSK-9 inhibitors remains mainly speculative and does not fit the conclusions.

Author Response

Dear Reviewer

Thank you for consideration and the opportunity to revise our manuscript. We highly appreciate the effort you dedicated to providing feedback with insightful and encouraging comments.

Our point-by-point response to the reviewer's comments are included in the attachment. Page and line numbers refer to the revised manuscript file with tracked changes. The bibliography has been updated accordingly.

We hope you find our responses satisfactory.

Sincerely,

Rahel Meier

Round 2

Reviewer 2 Report

The authors have adequately answered the questions.

The fact that only prescription is included in the FIRE database, but not dispensation or any other tracer of compliance, should be added to the manuscript as one additional aspect within the study´s limitations.

Author Response

Dear Reviewer

Thank you for consideration and the response to our revision. To comply with your request we added the following sentence to the limitation section (page: 11, lines 265 -267): “Additionally, it should be noted that the medication information stems from prescriptions, and information on patient compliance was not available.”

We hope you find our response satisfactory.

Sincerely,

Rahel Meier